# CCRANet: A Two-Stage Local Attention Network for Single-Frame Low-Resolution Infrared Small Target Detection

**Wenjing Wang** [1], **Chengwang Xiao** [1,*], **Haofeng Dou** [2], **Ruixiang Liang** [2], **Huaibin Yuan** [2], **Guanghui Zhao** [1], **Zhiwei Chen** [1] **and Yuhang Huang** [1]

1   Science and Technology on Multi-Spectral Information Processing Laboratory, School of Electronic Information and Communications, Huazhong University of Science and Technology, Wuhan 430074, China; m202072171@hust.edu.cn (W.W.); guanghui_zhao@hust.edu.cn (G.Z.); d201780536@hust.edu.cn (Z.C.); yuhang_huang@hust.edu.cn (Y.H.)

2   China Academy of Space Technology (Xi'an), Xi'an 710100, China; haofeng_dou@hust.edu.cn (H.D.); liangrx@cast504.com (R.L.); 201607046@xaau.edu.cn (H.Y.)

*   Correspondence: d201880571@hust.edu.cn

**Abstract:** Infrared small target detection technology is widely used in infrared search and tracking, infrared precision guidance, low and slow small aircraft detection, and other projects. Its detection ability is very important in terms of finding unknown targets as early as possible, warning in time, and allowing for enough response time for the security system. This paper combines the target characteristics of low-resolution infrared small target images and studies the infrared small target detection method under a complex background based on the attention mechanism. The main contents of this paper are as follows: (1) by sorting through and expanding the existing datasets, we construct a single-frame low-resolution infrared small target (SLR-IRST) dataset and evaluate the existing datasets on three aspects—target number, target category, and target size; (2) to improve the pixel-level metrics of low-resolution infrared small target detection, we propose a small target detection network with two stages and a corresponding method. Regarding the SLR-IRST dataset, the proposed method is superior to the existing methods in terms of pixel-level metrics and target-level metrics and has certain advantages in model processing speed.

**Keywords:** infrared image; small target detection; deep learning; self-attention

## 1. Introduction

Compared with visible light imaging detection and active radar imaging detection, infrared imaging detection technology has the following characteristics [1]: unaffected by light conditions, works in all types of weather, works passively, has high imaging spatial resolution, adapts to various environments, has strong anti-electromagnetic interference ability, has a simple structure, is small in size, and easy to carry and hide. Benefiting from the above advantages, infrared detection and imaging technology has been widely used in infrared search and tracking, infrared precise guidance, low and slow small aircraft detection and identification, and other projects [2].

In some cases that need to be pre-judged, the target to be detected is far away from the infrared detection imaging system, and the image shows a dim and small target, often lacking texture information. The targets to be detected are usually aircrafts, drones, missiles, ships, vehicles, and other fast-moving objects [3,4], so the outlines of the imaging targets are fuzzy. In addition, as they are affected by the surrounding environment and detection equipment, small infrared targets are easily submerged in noise and complex backgrounds [5]. All these factors bring challenges to infrared small target detection.

At present, there are many infrared detection devices with low imaging resolution that are applied in various fields [6]. Therefore, it is of practical significance to design a method for small target detection in low-resolution infrared images to improve the small target

detection performance. The number of pixels occupied by the target in the low-resolution infrared small target image is low [7], and a more accurate prediction of each pixel of the small target (that is, improving the pixel-level metrics of the low-resolution infrared image small target detection) can significantly improve the target detection performance.

Research on infrared small target detection is divided into single-frame image target detection and multi-frame image target detection [2]. This paper focuses on the former. Early researchers mainly proposed model-driven methods. Filter-based methods [8,9] require determining the filtering template in advance based on the structural characteristics of the image, so it has poor adaptability to complex background environments. Methods [10–13] based on local contrast are suitable for situations where there is a significant difference in grayscale between the target and surrounding background, but they are prone to missed detections and misjudgments. Low-rank-based [14,15] and tensor-based [16–18] methods can achieve good results, but the computational cost is high, and hyperparameters are more sensitive to image scenes.

With the development of deep learning, some data-driven methods and infrared small target datasets [7,19–22] have emerged in recent years. Considering the weak and small characteristics of infrared small targets, infrared small target detection is usually modeled as a semantic segmentation problem. In order to ensure that the features of small targets are not submerged, some methods [7,19,22,23] have been used to enhance the fusion of features at different layers of the network. Based on the small proportion of small targets in the overall image, some methods [24,25] solve the problem with infrared small target detection by suppressing the background area to make the network pay more attention to the target area. There are also some studies [26–29] that consider how to improve and innovate based on classic encoding and decoding structures.

The existing single-frame infrared small target detection methods [7,23,24] have problems in terms of poor adaptability and high false-alarm and missed detection rates when they are used to detect infrared small target images with low resolution. This is not only because the quality of the existing dataset is not high, which leads to unsatisfactory training of the network, but also related to the large number of parameters in the existing network structure or the insufficient local attention to small targets.

Therefore, in view of the problems in the existing datasets, we propose corresponding improvement strategies. Then, we construct a single-frame low-resolution infrared small target image dataset with high quality and a large amount of data called SLR-IRST. At the same time, we design a central point-guided circular region local self-attention network (CCRANet) for low-resolution infrared image small target detection. The CCRANet detects small targets in the image in two stages from coarse to fine. In the coarse stage, multiple circular regions of interest of fixed size are obtained by a center point-guided circular region of interest suggestion (CCRS) module. In the fine stage, the local feature information of small targets is further extracted in a share params local self-attention (SPSA) module, and each pixel is predicted. In the SLR-IRST dataset, the CCRANet can significantly improve the target detection effect on small targets. Compared with the present methods, the pixel-level metric, IoU, is improved by about 3%, the nIoU is improved by about 6%, and the target-level metric probability of detection ($P_d$) is improved by about 6%.

## 2. Related Works

### 2.1. Infrared Small Target Datasets

The Society of Photo-Optical Instrumentation Engineers (SPIE) defines infrared small targets as having a total spatial extent of less than 81 pixels (9 × 9) in a 256 × 256 image [30]—that is, the proportion of small targets in the entire image is less than 0.12%. In addition, the size of small infrared targets varies greatly, ranging from only one pixel (i.e., dot target) to dozens of pixels (i.e., expanded target) [29].

In recent years, some scholars have done a lot of work on the collection and production of infrared small target datasets and have publicly released these datasets, which include single-frame datasets [7,19–22] (see Table 1) and multi-frame datasets [31–33] (see Table 2).

**Table 1.** Details on the present single-frame infrared small target datasets.

| Data Type | Dataset | Image Num | Image Size | Provided Label | Target True Class | Background Type |
|---|---|---|---|---|---|---|
| Real | SIRST [19] | 427 | 96 × 135 to 388 × 418 | Pixel/Box | Aircraft/Drone/Ship/Vehicle | Cloud/Grass/River |
| | IRSTD-1k [20] | 1001 | 512 × 512 | Pixel | Drone/Bird/Animal | Cloud/Building/Grass/River/Lake |
| Synthetic | MFIRST [21] | 10,000 | 128 × 128 | Pixel | - | Cloud/Road |
| | NUDT-SIRST [7] | 1327 | 256 × 256 | Pixel | - | Cloud/Building/Vegetation |
| | IRST640 [22] | 1024 | 640 × 512 | Pixel | - | Cloud/Building |
| real/synthetic | SLR-IRST (our) | 2689 | 256 × 256 | Pixel/Box/Center | Aircraft/Drone/Ship/Vehicle/Bird/Animal | Cloud/Building/Grass/River/Lake/ Vegetation |

**Table 2.** Details on the present multi-frame infrared small target datasets.

| Data Type | Dataset | Sequence Num | Image Num | Image Size | Target True Class | Background Type |
|---|---|---|---|---|---|---|
| real | Dataset in [31] | 6 | 342 | 318 × 256 to 540 × 398 | Drone | City/Building/Tower Hanger |
| | ISATD [32] | 22 | 16177 | 256 × 256 | Drone | Sky/Field/Building |
| real/synthetic | IRDST [33] | 401 | 142727 | 720 × 480/ 934 × 696 | Aircraft/Drone | Clouds/Trees/Lakes/Buildings |

In Tables 1 and 2, it can be seen that the sample size of real single-frame infrared small target data is relatively small, but the sample size of multi-frame infrared small target data is rich, which can be used to expand single-frame data. Constructing a single-frame infrared small target dataset with a larger data volume and higher quality can promote the development of single-frame infrared small target detection.

### 2.2. Infrared Small Target Detection Methods

In recent years, deep learning has developed rapidly in terms of solving visual tasks such as image classification, object detection, and semantic segmentation. Some methods based on deep learning have also emerged for infrared small target detection.

Due to their "weak" and "small" characteristics, infrared small targets are easily overwhelmed by a network's high-level features. However, if only low-level features are used, it is not possible to fully comprehend semantic information, making it easy to miss detection and raise false alarms. Therefore, some researchers have combined attention mechanisms to study methods for enhancing feature fusion at different layers. Dai et al. proposed a bottom-up channel attention modulation method [23] (ACM) to preserve and highlight infrared small target features in high-level layers. Thereafter, Dai et al. [19] modularized local contrast measurement methods [10] from traditional methods in the network to design a model-driven deep learning network (ALCNet). Li et al. [7] proposed the use of DNANet to achieve progressive information interaction between high-level and low-level features through densely nested modules (DNIM). Chen et al. [22] introduced the self-attention mechanism of a transformer into the designed IRSTFormer to extract multi-scale features from the input image through the overlapping block self-attention structure of the hierarchy.

Based on the small proportion of small targets in the overall image, some researchers solve the problem with infrared small target detection by suppressing the background area so that the network pays more attention to the target area. Wang et al. [24] proposed a coarse-fine two-stage network, IAANet. In the coarse stage, the candidate target regions are obtained by the region proposal network (RPN), and in the fine stage, the global features of all candidate target regions in the image are extracted by the attention encoder (AE). In IAANet, the hard decision method is used to suppress the background regions to the greatest extent. Cheng et al. [25] designed a supervised attention module trained by small target diffusion maps in the proposed LPNet to suppress most of the background pixels irrelevant to small target features in a soft decision manner.

It has been proved that the classical encoder-decoder structure can achieve better results in the semantic segmentation task [34], and some researchers have carried out

work on the improvement and innovation of the classical codec structure. Tong et al. [26] proposed MSAFFNet, which introduced the EIFAM module containing edge information based on the codec structure and constructed multi-scale labels to focus on the details of target contour and internal features. Wu et al. [27] proposed UIU-Net (U-Net in U-Net). It imparts a tiny U-Net into a larger U-Net backbone network to realize multi-level and multi-scale representation learning of objects. Chen et al. [28] proposed a MultiTask-UNet (MTUNet) with both detection and segmentation heads. By sharing the backbone, the similar semantic features of the two tasks are fully utilized. Compared with the compound single-task model, MTUNet has fewer parameters and faster inference speed. Wu et al. [29] proposed a multi-level TransUNet (MTU-Net). In the encoder, the features extracted by convolution are passed through a multi-level ViT module (MVTM) to capture remote dependencies.

The networks that combine the attention mechanism and multi-scale feature fusion [7,19,22,23] enhance the network's ability to extract image features, but the local attention to small targets in the image is not enough and the network's ability to detect small targets is not enough to be improved. The networks that focus on the localized region of small targets [24,25] have a problem: the number of network parameters and computational amount are larger, and the network prediction speed is slower.

## 2.3. Evaluation Metrics

The output of the infrared small target detection network is pixel-level segmentation. Therefore, it is common to use semantic segmentation metrics to evaluate network performance, such as precision, recall, *F1* score, ROC curve, and PR curve.

Precision and recall refer to the proportion of correctly predicted positive samples out of all predicted positive samples and all true positive samples, respectively. The *F1*-Score is the harmonic mean of precision and recall. The definitions of *P* (precision), *R* (recall), and *F1* − *P* (F1 score) are as follows:

$$
\begin{aligned}
P &= \frac{TP}{TP+FP} \\
R &= \frac{TP}{TP+FN} \\
F1 - P &= \frac{2P \cdot R}{P+R}
\end{aligned}
\tag{1}
$$

where $T$, $P$, $TP$, $FP$, and $FN$ denote the true, positive, true positive, false positive, and false negative, respectively.

The receiver operating characteristic (ROC) curve shows how the model performs across all classification thresholds. The horizontal coordinate of the ROC curve is the false positive rate (FPR), and the vertical coordinate is the true positive rate (TPR). It goes through the points (0, 0) and (1, 1). The horizontal coordinate of the precision-recall (PR) curve is the recall rate, which reflects the classifier's ability to cover positive examples. The vertical coordinate is the precision, which reflects the accuracy of the classifier's prediction of positive examples.

However, as a target detection task, some researchers have proposed pixel-level and target-level evaluation metrics based on existing metrics to better evaluate the detection performance of infrared small targets.

IoU and nIoU are pixel-level metrics. IoU represents the ratio of intersection and union between the predicted and true results:

$$
\text{IoU} = \frac{TP}{T + P - TP}
\tag{2}
$$

nIoU [19] is the numerical result normalized by the IoU value of each target, as shown in (3), where $N$ represents the total number of targets.

$$
\text{nIoU} = \frac{1}{N} \sum_{i}^{N} \frac{TP[i]}{T[i] + P[i] - TP[i]}
\tag{3}
$$

$P_d$ (probability of detection) and $F_a$ (false-alarm rate) are target-level metrics [22]. $P_d$ measures the ratio of the number of correctly predicted targets to the number of all targets. $F_a$ measures the ratio of incorrectly predicted pixels to all pixels in the image.

$$P_d = \frac{\text{\# num of true detections}}{\text{\# num of actual targets}} \tag{4}$$

$$F_a = \frac{\text{\# num of false predicted pixels}}{\text{\# num of all pixels}} \tag{5}$$

## 3. SLR-IRST Dataset

This section introduces the construction process and statistical characteristics of the single-frame low-resolution infrared small target image dataset (SLR-IRST).

### 3.1. Construction of the SLR-IRST Dataset

Unlike traditional model-driven methods, deep learning is a data-driven method. It requires a large amount of diverse data training to improve the generalization ability of the network [35]. At present, few real single-frame infrared small target images have been published publicly [19,20], and the resolution of the images is inconsistent. The published synthetic single-frame infrared small target images have high similarity and low synthesis quality [21,22]. All these problems affect the training coupling process and the final performance of the network [7]. Therefore, we constructed a single-frame low-resolution infrared small target image dataset (SLR-IRST) based on the existing infrared small target datasets and other infrared datasets through data collation and data expansion. The generation process of the SLR-IRST dataset is as follows (as shown in Figure 1).

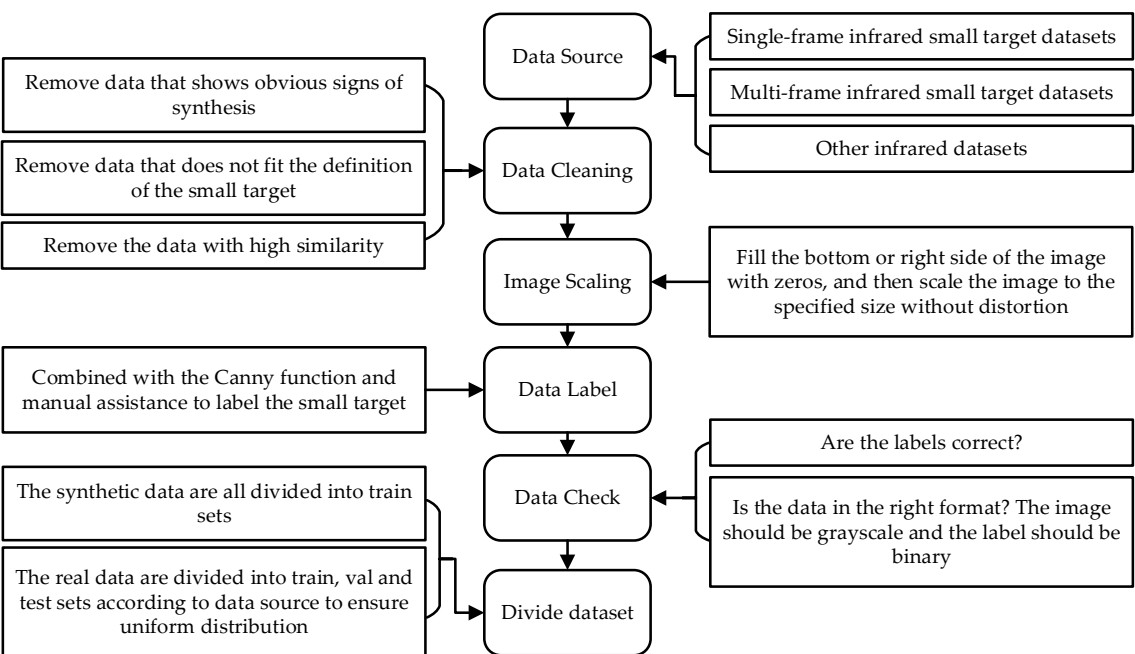

**Figure 1.** Construction process of the SLR-IRST dataset.

1. Data Source: The data in the SLR-IRST dataset are from a single-frame infrared small target dataset (SIRST, IRSTD-1k, IRST640, SIRST-v2 [36]) and a multi-frame infrared small target dataset (IRDST [25]). There is a serious shortage of maritime target data in the existing dataset (see Table 3), so some waterborne target images were extended in the SLR-IRST dataset from this infrared dataset (http://iray.iraytek.com:7813/apply/Sea_shipping.html/ (accessed on 10 December 2021)).

**Table 3.** Real target class statistics of the SLR-IRST dataset and other mainstream infrared small target datasets.

| Dataset | Airborne Target | Waterborne Target | Ground Target |
|---|---|---|---|
| SIRST | 394 | 15 | 18 |
| IRST640 | 1024 | 0 | 0 |
| IRSTD-1K | 516 | 87 | 398 |
| SLR-IRST | 1960 | 312 | 417 |

2.  Data Cleaning: The collected images were first data cleaned. The images that did not meet the small target definition [30] or had obvious synthetic traces were removed. The frequency of image extraction in the multi-frame dataset, IRDST, is 1 for every 50 images. In order to ensure the diversity and balance of the data in the dataset, the structure similarity index measure (SSIM) value [37] was used to evaluate the similarity between the images. Highly similar images with SSIM values greater than 0.85 (for simple backgrounds) and 0.90 (for complex backgrounds) were discarded.

3.  Image Scaling: The resolution of the images in the existing infrared small target dataset is inconsistent. Under normal circumstances, the images in the dataset would be directly resized to the specified resolution before training and testing [19]. However, the direct resize operation causes the target to deform and the label to no longer be binarized, which introduces additional errors when network testing. Therefore, the resolution of images in SLR-IRST was unified by an undistorted method [38]. Zero was filled below or to the right of the original image to make it match the aspect ratio of the specified resolution, and then resized to the specified resolution. By referring to the common resolution of infrared detection equipment and parameters of other datasets [32], the unified resolution of images in SLR-IRST was $256 \times 256$.

4.  Data Label: The images, after the unified resolution, were re-labeled. The small target in the infrared image was fuzzy and the edge was difficult to define. The Canny function [39] helps to define the boundaries of small targets and reduce manual labeling errors. The SLR-IRST dataset has pixel-level labels, bounding box labels, and center point labels. The boundary box label and center point label are determined by calculating the boundary and center point of the pixel-level label.

5.  Data Check: Finally, all the marked data was checked again to ensure that the labels are correct and the data format is correct.

6.  Divide Dataset: When dividing the dataset, all the synthesized data was divided in the training set. All real data was evenly divided into training, validation, and test sets depending on the source of the data.

### 3.2. Construction and Characterization of the SLR-IRST Dataset

Table 4 presents some basic statistics of the SLR-IRST dataset and other mainstream infrared small target datasets. In addition, all single-frame infrared small target datasets are evaluated in terms of target number, target category, target size, etc.

**Table 4.** Basic statistics of the SLR-IRST dataset and other mainstream infrared small target datasets.

| Dataset | SLR-IRST | SIRST | IRST640 | IRSTD-1K |
|---|---|---|---|---|
| image size | $256 \times 256$ | $96 \times 135$ to $388 \times 418$ | $640 \times 512$ | $512 \times 512$ |
| image num | 2689 | 427 | 1024 | 1001 |
| target num | 3586 | 533 | 1662 | 1495 |
| target pixel range | 1~367 | 4~330 | 1~51 | 1~1065 |
| (average) | (15.48) | (32.86) | (27.73) | (50.11) |
| target size range | $1 \times 1$~$14 \times 34$ | $2 \times 2$~$14 \times 34$ | $1 \times 1$~$9 \times 8$ | $1 \times 1$~$56 \times 53$ |
| (average) | ($3.33 \times 4.58$) | ($5.62 \times 6.94$) | ($6.11 \times 6.33$) | ($7.69 \times 8.74$) |
| target SCR range | 0.004~70.35 | 0.17~42.81 | 0.04~70.34 | 0.004~68.76 |
| (average) | (7.46) | (9.20) | (4.93) | (7.13) |

The SLR-IRST dataset has the following characteristics.

1.  More data: In Table 4, it can be seen that the SLR-IRST dataset has a much larger amount of data (2689 images, 3586 targets). As can be seen in Table 1, the scene types and target types in the SLR-IRST dataset are more abundant.

2.  Smaller target: As can be seen in Table 4, targets in the SLR-IRST dataset are smaller, occupying only 15.48 pixels on average and sized 3.33 × 4.58 on average. As can be seen in Figure 2a, compared with other datasets, more targets in the SLR-IRST dataset accounted for less than 0.005% of the total image.

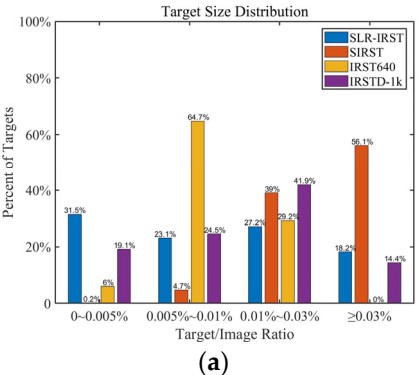 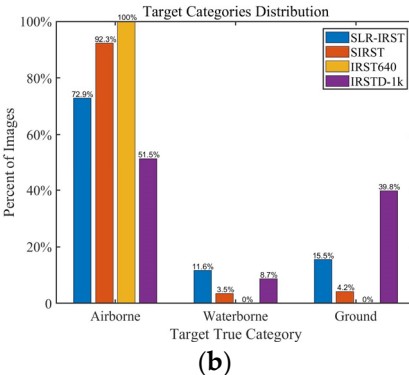 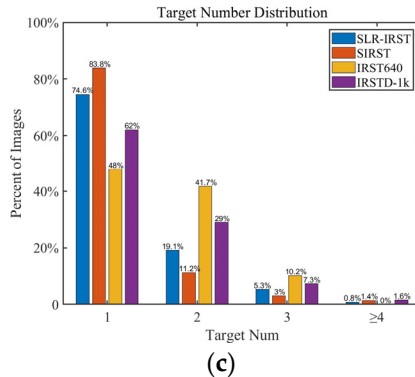

| (a) | (b) | (c) |

**Figure 2.** Statistical properties of SLR-IRST. (**a**) Statistical properties of target quantity distribution. (**b**) Statistical properties of target class distribution. (**c**) Statistical properties of target size distribution.

3.  More balanced distribution of data categories: Figure 2b shows the statistical distribution of targets in different datasets against the real categories. The distribution of target categories in the SLR-IRST dataset is more balanced. The sample size of the waterborne targets was expanded from 87 (8.7%) in IRSTD-1k to 313 (11.3%) in SLR-IRST, which is a nearly three-fold increase. This greatly alleviates the problem of sample scarcity in the waterborne target category.

In addition, as can be seen in Figure 2c, images with no more than 3 small targets account for more than 98% of all datasets. It shows that the small target in the infrared image has sparsity. As can be seen in Figure 2a, the small target in the infrared image has more targets than 0.03% of the image. It indicates that the small target in the infrared image is very small.

The SCR value [14] is the normalized value of the difference between the gray value of the target and the surrounding background area. The SCR value can be used to describe the difficulty of infrared small target detection. The larger the SCR value, the easier the target is to detect, and the smaller the SCR value, the harder it is to detect. As seen in Table 4, the average SCR value across all datasets is less than 10. It indicates that there is a large number of weak targets that are difficult to detect in the infrared small target dataset—that is, the small targets in the infrared image are weak.

We selected state-of-the-art (SOTA) networks, DNANet and ALCNet, and compared their IoU metric results in the SLR-IRST dataset and the SIRST dataset. As shown in Table 5, both DNANet and ALCNet showed a large degree of performance degradation in the SLR-IRST dataset. This shows that the SLR-IRST dataset presents a new challenge. It is necessary to design a network according to the characteristics of low-resolution infrared small target images to improve the detection ability of the network.

**Table 5.** Comparison of IoU metrics between the SOTA networks in the SIRST and SLR-IRST datasets.

| Method | SIRST | SLR-IRST |
|--------|-------|----------|
| ALCNet | 0.7570 | 0.7077 |
| DNANet | 0.7757 | 0.7076 |

## 4. Methodology

This section introduces specific information on the proposed CCRANet. First, Section 4.1 introduces the overall architecture of the CCRANet. Then, Section 4.2–4.4 introduce the specific structure of the network in detail.

### 4.1. Overall Architecture

Small targets in low-resolution infrared images are small in size, faint, and sparse. Combining these characteristics, we designed a central point-guided circular region local self-attention network (CCRANet) suitable for low-resolution infrared small target detection.

As shown in Figure 3, the CCRANet consists of a center point-guided circular region of interest suggestion (CCRS) module, a U-shaped feature extraction (U-FE) module, and a share params Local self-attention (SPSA) module. The entire working process is as follows: for the input image, $X \in \mathbb{R}^{H \times W}$, firstly, multiple central points of interest in the image are generated by the CCRS module, and the position information $b$ of the circular regions of interest guided by the central points is generated. The non-interest region is the output as the background in the prediction result. At the same time, feature extraction is carried out on the input image through the U-FE module to obtain the feature map, $\hat{X}$. By combining $b$ and $\hat{X}$, the local feature information, $x$, of multiple regions of interest is obtained. It will be fed into the SPSA module for self-attention calculations within each local region to obtain the feature map, $\hat{x}$. Finally, the feature map, $\hat{x}$, is used to predict the regions of interest using multi-layer perceptron (MLP). The three modules in the network are further described below.

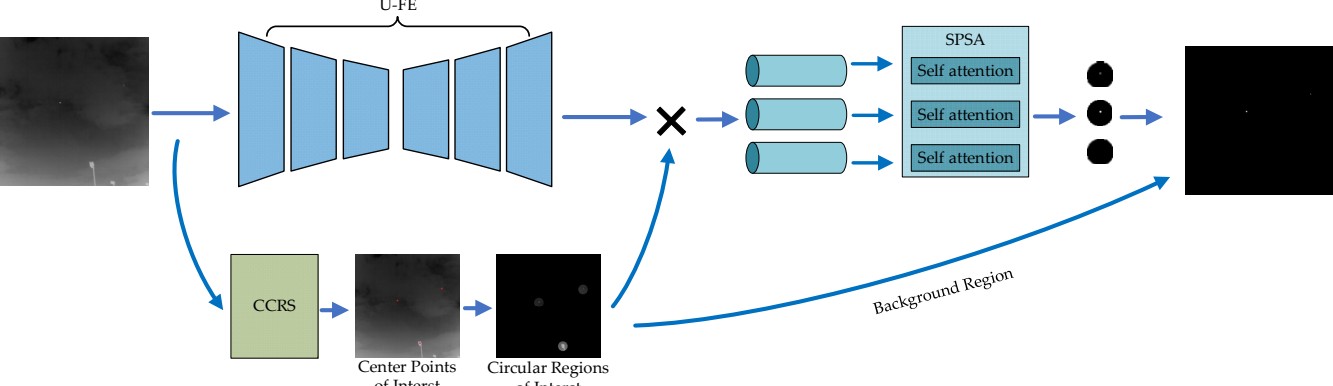

**Figure 3.** Overall architecture of the CCRANet.

### 4.2. Center Point-Guided Circular Region of Interest Suggestion Module

The small target in the infrared image only accounts for a very small part of the image, and most of the regions in the infrared image are redundant and interference information. At the same time, because of the long detection distance, the shapes of small targets in infrared images are almost round. Therefore, we designed a center point-guided circular region of interest suggestion (CCRS) module to obtain multiple centers of interest points in the image through a simple network. Under the guidance of these central points, multiple circular regions of interest are generated. Regions of non-interest are directly considered as the background.

In CCRS, the input image is progressively reduced by multiple convolutional layers (MCL) to obtain the feature map, $x_d \in \mathbb{R}^{h \times w \times c}$. $x_d$ has a step size of 16 relative to the input image. As shown in Figure 4, MCL has a similar structure to ResNet18 [40] and uses ResNet18's pre-training weight information. Then, after a convolution with a kernel size of one, the number of channels of $x_d$ is reduced to three. Let $(c_x, c_y)$ represent the spatial position of a point in the feature map, $x_d$. $s$ represents the subsampling multiple of the feature map, $x_d$, corresponding to the original image. $t_x$, $t_y$, and $t_o$ represent the three

parameters corresponding to the three channels in the feature map, $x_d$. The calculation method for the location information and confidence of the center point predicted by the network is as follows:

$$\hat{c}_x = (2\sigma(t_x) - 0.5 + c_x) \times s$$
$$\hat{c}_y = (2\sigma(t_y) - 0.5 + c_y) \times s \qquad (6)$$
$$\hat{o} = \sigma(t_o)$$

where $\hat{c}$ represents the predicted center point. $(\hat{c}_x, \hat{c}_y)$ and $\hat{o}$ represent the coordinates and confidence of the predicted central point, respectively. $\sigma$ stands for the sigmoid function. For an input image with a resolution of $256 \times 256$, 256 predicted centers of interest will be generated.

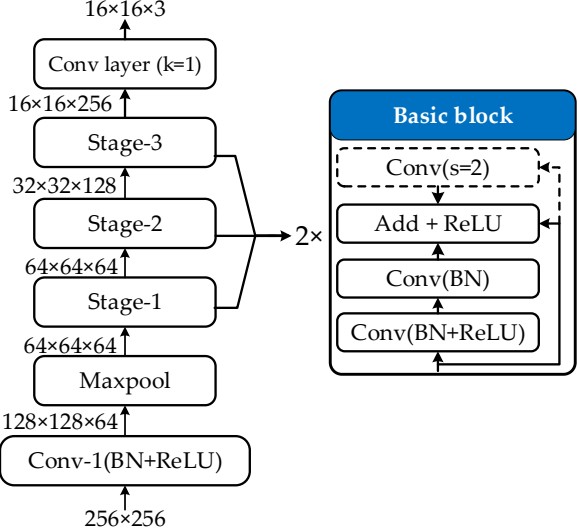

**Figure 4.** Architecture of the CCRS module. The dashed part appears only in the first basic block in stage 2 and stage 3.

In the CCRS module, there is a confidence threshold, *conf*. The center point with confidence above *conf* is considered as the interest center point that is needed. At the beginning of training, the number of retained centers of interest is variable. It may be the case that a large number of central points are retained or that no central points are retained. So, the CCRS module sets an additional hyperparameter, *k*, which represents the maximum number of retained center points of interest. When the number of center points higher than *conf* exceeds *k* or is 0, the first *k* center points with higher confidence are retained. In addition, there is another hyperparameter, *l*, which is the radius of the circular region of interest generated based on the central point guidance. The central coordinate information of the circular regions of interest generated by the CCRS is expressed as $b_{box} \in \mathbb{R}^{n \times 2}$ $(0 \leq n \leq k)$.

### 4.3. U-Shape Feature Extraction Module

The U-shape feature extraction (U-FE) module is parallel to the CCRS module, and its overall structure is U-shaped, which is similar to that of the U-Net network [34]. In the U-FE module, the downsampling times are reduced to two times to prevent the feature information of the small target being drowned in the deep feature. The downsampling operation is implemented using MaxPool, and the upsampling operation is implemented using ConvTranspose. As shown in Figure 5, for the input infrared image, $X \in \mathbb{R}^{H \times W}$, after passing through the U-shaped subnetwork, the feature map, $\hat{X} \in \mathbb{R}^{H \times W \times D}$, is obtained. Among them, $D = 64$.

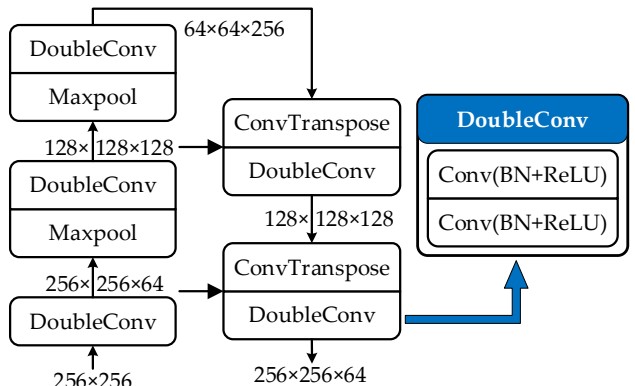

**Figure 5.** U-FE module architecture. DoubleConv consists of two $3 \times 3$ convolutions (including BN and ReLU).

### 4.4. Share Params Local Self-Attention Module

According to the output, $b_{box} \in \mathbb{R}^{n \times 2}$ and $\hat{X} \in \mathbb{R}^{H \times W \times D}$, of the first two stages, the characteristic information, $x_i \in \mathbb{R}^{\pi l^2 \times D} (i = 1, 2, \ldots, n)$, of each circular region of interest is obtained first (see Figure 6). Then, all $x_i$ is spliced together along the new dimension to obtain $x \in \mathbb{R}^{k \times \pi l^2 \times D}$, which is the feature information of all circular regions of interest in the image. (When $n < k$, the data size is kept consistent by filling 0 ($x_0 \in \mathrm{O}^{\pi l^2 \times D}$)).

$$x = \mathrm{concat}(\underbrace{x_1 + x_2 + \ldots + x_n + x_0 + \ldots}_{k}) \ (0 < n \le k) \tag{7}$$

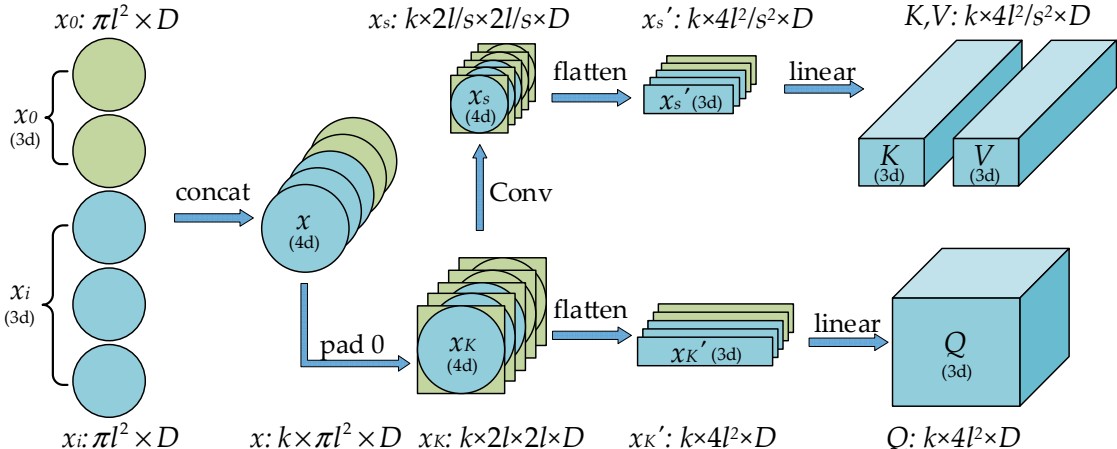

**Figure 6.** Each step in the SPSA from the circular region of interest features to the matrices *Q*, *K*, *V* in self-attention. (*n*d) means that the feature matrix is *n*-dimensional (*n* = 2, 3, 4).

The SPSA module uses the self-attention mechanism to further extract features from the regions of interest because the self-attention mechanism has a global receptive field and is better able to capture the internal correlation between features. The traditional self-attention computing process is proportional to the square of the length of the input feature sequence [41,42]. Considering that there are many feature sequences for self-attention calculation, the calculation amount is large. The SPSA adds scaling down operations to the traditional multi-head attention layer (MHA).

As shown in Figure 6, the feature sequence, *x*, is first restored to the form of a two-dimensional feature map along the spatial dimension. Since the region of interest is circular, when restoring along the spatial dimension, 0 is added around the circular region to make it square, and the feature map, $x_K \in \mathbb{R}^{k \times 2l \times 2l \times D}$, is obtained. The scaling operation is carried

out in the spatial dimension through a layer of convolution. Let the scaling reduction factor be $s$—that is, the kernel size and step size of the convolution are both $s$. The feature map, $x_s \in \mathbb{R}^{k \times 2l/s \times 2l/s \times D}$, is obtained by scaling down the feature map, $x_K$. The feature sequence, $x'_K \in \mathbb{R}^{k \times N \times D}$ $(N = 4l^2)$, $x'_s \in \mathbb{R}^{k \times N' \times D}$ $(N' = 4l^2/s^2)$, is obtained by flattening $x_K$, $x_s$ along the spatial dimension. The keyword ($K$) and the value ($V$) are generated from the scaled feature sequence, $x'_s$, through two sets of linear projections. The query ($Q$) is generated from the unscaled feature sequence, $x'$, through a set of linear projections.

$$Q = \text{Linear}(\text{Flatten}(x_K))$$
$$K = \text{Linear}(\text{Flatten}(\text{Conv}_s(x_K))) \qquad (8)$$
$$V = \text{Linear}(\text{Flatten}(\text{Conv}_s(x_K)))$$

where, Flatten represents flattening feature maps along the spatial dimension, $\text{Conv}_s$ represents a convolution of both kernel size and step size with $s$, and Linear represents a linear layer.

As shown in Figure 7, the calculation of multi-head self-attention is as follows:

$$\text{MultiHead}(Q, K, V) = \text{concat}(\text{head}_1, \ldots, \text{head}_h)W^O$$
$$\text{where head}_i = \text{Attention}(Q_i, K_i, V_i) = \text{softmax}\left(\frac{Q_i K_i^T}{\sqrt{d_k}}\right)V_i \qquad (9)$$

and where $W^O \in \mathbb{R}^{D \times D}$, $Q_i \in \mathbb{R}^{k \times N \times d}$, $K_i \in \mathbb{R}^{k \times N' \times d}$, $V_i \in \mathbb{R}^{k \times N' \times d}$, and $d = D/h$. $h$ represents the number of heads. The scaling operation does not change the length and dimension of the output feature sequence but only reduces the computation amount in the process of self-attention calculation.

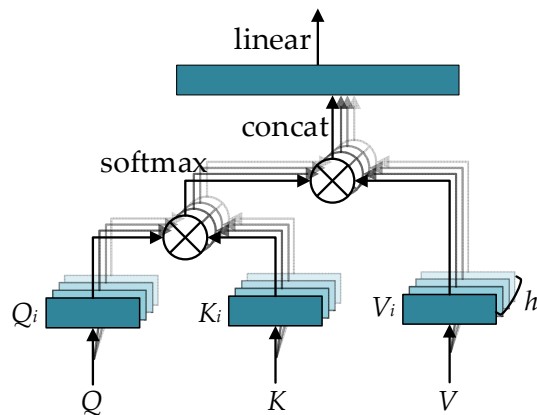

**Figure 7.** Architecture of the multi-head self-attention mechanism for scaling down in the SPSA.

Other visual task algorithms [24,43,44] usually splice the features in the same image along the spatial dimension because it is believed that the features in the same image are related and influence each other. It is worth noting that in the SPSA, feature splicing of multiple regions of interest in the same image is performed in a new dimension. In this way, self-attention calculation is only confined to the interior of each region of interest, and no self-attention calculation is performed among features of different regions. We believe that the infrared small target task has different characteristics from other vision tasks. The infrared small targets in images are small and sparse and have obvious locality. The features of other regions in the image have little relevance to the prediction of small targets and are redundant information. Therefore, it is unnecessary to concatenate the features in the infrared small target image along the spatial dimension, as it will increase the amount of calculation. In addition, in the SPSA, the process of calculating self-attention for different regions of interest is parallel, and the parameters are shared. Since there is no need to calculate the self-attention between different areas of interest, the amount of

computation and the number of parameters in the process of self-attention calculation are greatly reduced.

For $k$ input regions of interest with radius $l$ and number of channels $D$, the computational complexity of the calculation using the SPSA, or the standard multi-head self-attention mechanism (*TE*) are as follows:

$$\Omega(TE) = 2k^2\pi^2 l^4 D + 4k\pi l^2 D^2 \tag{10}$$

$$\Omega(SPSA) = \frac{8kl^4 D}{s^2} + 4kl^2 D^2 s^2 \tag{11}$$

Of the two items that make up computational complexity, the former contributes the vast majority of the computational effort. Using the SPSA for self-attention computation can greatly reduce the computation of the former and speed up the reasoning speed of the network.

## 5. Experiments

In this section, the experimental setup is introduced first, including implementation details and evaluation metrics. Then, the ablation experiments on the loss function, hyperparameter setting, and network structure design are shown to verify the rationality of the network design. Then, the proposed CCRANet is compared with other state-of-the-art (SOTA) methods to verify the effectiveness of the network. Finally, the CCRANet and other SOTA methods are compared in actually collected infrared small target images to show that the network has better generalization ability.

### 5.1. Experimental Setting

#### 5.1.1. Implementation Details

In the network training stage, the optimizer is the adaptive gradient (AdaGrad) [45] method, and the method to initialize the weight and bias of the model is Xavier [46]. The CCRS module has a learning rate of 0.01, and the rest of the network has a learning rate of 0.001. The first epoch of network training trains only the CCRS module. The batch size and epoch are set to 16 and 500, respectively. All networks rely on PyTorch [47]. The computer used in the experiment has an Intel(R) Core (TM) i9-10920X @ 3.50 GHz CPU and a Nvidia RTX 3090 GPU. All experiments were trained and tested on the SLR-IRST dataset, and the number of images in the training set, verification set, and test set were 1616, 533, and 540, respectively, with a partition ratio of about 6:2:2. The SLR-IRST dataset and CCRANet will be made public on https://github.com/wangwjinggg/CCRANet.

Before the training began, data enhancement was performed on the training data. The input images were scaled at a random scale. The aspect ratio of the scaled image ranged from 0.58 to 1.85. The length of the longer side of the scaled image ranged from 0.5 to 2 times the original width and height. The scaled image was uniformly and arbitrarily cropped to a $256 \times 256$ size. In addition to this, data enhancement strategies also include random horizontal flipping and random superimposed Gaussian noise.

#### 5.1.2. Evaluation Metrics

Infrared small target detection networks based on deep learning [7] usually use both pixel-level and target-level metrics to comprehensively evaluate the performance of the network.

Previously, the false-alarm rate ($F_a$) of the target-level metrics (see Formula (5)) represented the ratio of the wrongly predicted pixels to all pixels of the image and did not really measure the false-alarm situation of the network from the target-level. Therefore, we redefined the $F_a$ as follows:

$$F_a = \frac{\text{\# num of false detections}}{\text{\# num of predicted targets}} \tag{12}$$

In addition, we also defined the $M_d$ (miss detection rate), the $P_a$ (probability of accuracy), and the $F1 - T$ (F1 score) of the target-level evaluation metrics:

$$M_d = \frac{\# \text{ num of not detections}}{\# \text{ num of actual targets}} \tag{13}$$

$$P_a = \frac{\# \text{ num of true detections}}{\# \text{ num of predicted targets}} \tag{14}$$

$$F1 - T = \frac{2P_d \cdot P_t}{P_d + P_t} \tag{15}$$

where $P_d$ represents the probability of detection (see Formula (4)). When the center of the predicted target is within $d$ pixels of the center of the real target, the target is considered to be correctly detected. The threshold $d$ is set to two in this paper. The $F1 - T$ takes both recall and accuracy into account to measure target-level prediction. A method is considered good when high values are obtained on $P_d$ (↑), $P_a$ (↑), $F1 - T$ (↑), and low values are obtained on $M_d$ (↓), $F_a$ (↓).

The pixel-level metrics used for evaluation include IoU, nIoU, $R$ (Recall), $P$ (Precision), and $F1 - P$ scores. In addition, the receiver operating characteristic (ROC) and precision-recall (*PR*) curves are used to further demonstrate the overall detection effect, target detection capability, and background suppression capability.

### 5.2. Loss Function Experiments

Loss in the CCRANet consists of two parts including the loss of regression learning from the predicted center point in the CCRS module and the loss of regression learning from the predicted pixel-level segmentation results. The regression learning of the predicted center point includes two parts: location information learning and confidence learning. The BCEWithLogits loss function is used to monitor the learning of confidence and the L1 loss function to monitor the learning of location information. The metrics results for monitoring learning of pixel-level segmentation results with four different loss functions are compared in this section, including BCEWithLogits, Soft-IoU [19], Focal [48], and Mixed. The Mixed loss function is defined as follows:

$$\mathcal{L}_{Mixed} = \mathcal{L}_{\text{BCEWithLogits}} + \mathcal{L}_{\text{Soft-IoU}} \tag{16}$$

In order to suppress the imbalance of positive and negative samples, the balance coefficient is set when using the BCEWithLogits loss function. Among them, the balance coefficient is set to $a = 10$ in central point regression learning and $a = 3$ in semantic segmentation learning.

It can be seen from Table 6 that BCE obtained better index results. It shows that the BCEWithLogits loss function, after the balanced adjustment of positive and negative sample proportions, has a better monitoring ability in terms of binary classification tasks. Based on the above experimental results, the loss function used in the CCRANet training is as follows:

$$\mathcal{L} = (\mathcal{L}_{\text{Mixed}})_{\text{for seg}} + 0.5 \times (\mathcal{L}_{L1} + \mathcal{L}_{\text{BCEWithLogits}})_{\text{for center}} \tag{17}$$

**Table 6.** Comparison of pixel-level metrics for networks trained with different loss functions. The best result values are marked in **bold**.

| Seg Loss | Center Loss | nIoU (↑) | IoU (↑) | R (↑) | P (↑) | F1 − P (↑) |
|----------|-------------|----------|---------|-------|-------|------------|
| BCE | L1+BCE | 0.7340 | 0.7286 | **0.8562** | 0.8275 | 0.8416 |
| Focal | L1+BCE | 0.7398 | 0.7216 | 0.7988 | **0.8703** | 0.8330 |
| Soft-IoU | L1+BCE | 0.7359 | 0.7178 | 0.8069 | 0.8366 | 0.8215 |
| Mixed | L1+BCE | **0.7579** | **0.7398** | 0.8324 | 0.8694 | **0.8505** |

### 5.3. Hyperparameter Experiments

This section conducts comparative experiments on the values of the two hyperparameters $k$ and $l$ in the CCRS module, and the experimental results are compared in Table 7.

**Table 7.** Comparison of pixel-level and target-level metrics of the CCRS module hyperparameters under different settings. The best result values are marked in **bold**.

| $k$ | $l$ | Pixel-Level Metric | | | | | | Target-Level Metric | | | |
|---|---|---|---|---|---|---|---|---|---|---|---|
| | | nIoU ($\uparrow$) | IoU ($\uparrow$) | $R$ ($\uparrow$) | $P$ ($\uparrow$) | $F1 - P$ ($\uparrow$) | $P_d$ ($\uparrow$) | $M_d$ ($\downarrow$) | $P_a$ ($\uparrow$) | $F_a$ ($\downarrow$) | $F1 - T$ ($\uparrow$) |
| 3 | 10 | 0.7348 | 0.7271 | 0.8682 | 0.8119 | 0.8391 | 0.9587 | 0.0413 | 0.8256 | 0.1744 | 0.8872 |
| 10 | 10 | 0.6764 | 0.6941 | **0.8853** | 0.7498 | 0.8119 | 0.9516 | 0.0484 | 0.7286 | 0.2714 | 0.8253 |
| 5 | 10 | **0.7579** | **0.7398** | 0.8324 | **0.8694** | **0.8505** | **0.9659** | **0.0341** | **0.8773** | **0.1227** | **0.9194** |
| 5 | 5 | 0.7216 | 0.6658 | 0.7681 | 0.8287 | 0.7973 | 0.9516 | 0.0484 | 0.8253 | 0.1747 | 0.8840 |
| 5 | 20 | 0.7395 | 0.7287 | 0.8234 | 0.8296 | 0.8265 | 0.9559 | 0.0441 | 0.7936 | 0.2064 | 0.8672 |

It can be seen that the metrics results are the best when $k = 5$ and $l = 10$. When $k = 3$, the metrics drop slightly, and when $k = 10$, they drop sharply. This is because when the number of circular regions of interest is too large, some background regions containing suspected target characteristics are also preserved. This introduces interference information, which causes the precision ($P$) and $P_a$ of the network to decrease significantly. Metrics at $l = 5$ and $l = 20$ both decreased compared with those at $l = 10$. This indicates that when the circular region of interest is too small, some small targets are not fully wrapped, resulting in a significant decrease in the network pixel-level recall rate ($R$) metric. When the circular region of interest is too large, it contains more interference information, resulting in a significant decrease in the target-level probability of accuracy ($P_a$) metric of the network. To sum up, the values of the two hyperparameters in the CCRS module are set to $k = 5$ and $l = 10$.

### 5.4. Ablation Study

Some ablation studies were conducted to verify the rationality and validity of the CCRANet in terms of network structure. As shown in Table 8, the proposed CCRANet is used as the benchmark network, denoted as A. Based on A, the U-FE module is replaced with the FPN [49] structure, denoted as B1. The number of feature map channels output in B1 is set to 64, which is consistent with that in A. On the basis of A, the U-FE module is replaced with a 7-layer fully convolution FCN structure without downsampling, denoted as B2. The 7-layer convolution gradually increases the number of channels of the feature to 256, then maintains the 3-layer convolution, and then gradually reduces the number of channels to 64, which is consistent with the output in A.

**Table 8.** Network settings for the ablation experiment.

| CCRS | U-FE | FPN | FCN | SPSA | TSA | CONV | Name |
|---|---|---|---|---|---|---|---|
| $\checkmark$ | $\checkmark$ | | | $\checkmark$ | | | A (CCRANet) |
| $\checkmark$ | | $\checkmark$ | | $\checkmark$ | | | B1 |
| $\checkmark$ | | | $\checkmark$ | $\checkmark$ | | | B2 |
| $\checkmark$ | $\checkmark$ | | | | $\checkmark$ | | C1 |
| $\checkmark$ | $\checkmark$ | | | | | $\checkmark$ | C2 |

Based on A, the SPSA module was replaced with traditional self-attention (TSA) [41,42], denoted as C1. In TSA, the feature sequences of multiple circular regions of interest are concatenated along spatial dimensions. Moreover, there is no scaling down operation through convolution in the multi-head self-attention of TSA. The self-attention calculation in TSA is performed among the features of all the circular regions of interest in the image. Based on A, the SPSA module was replaced with a 4-layer convolution (CONV) structure, denoted as C2. The CONV convolution has a kernel size of $3 \times 3$, step size of 2, and

padding of 1 and does not change the number of channels of the original feature map. In keeping with the residual connection structure in the SPSA, a residual link is also added here for each layer of convolution. The metrics of the ablation experiment are shown in Table 9.

**Table 9.** Comparison of pixel-level and target-level metrics for the ablation experiments. The best result values are marked in **bold**.

| Method | Pixel-Level Metric | | | | | Target-Level Metric | | | | |
|---|---|---|---|---|---|---|---|---|---|---|
| | nIoU (↑) | IoU (↑) | R (↑) | P (↑) | F1 − P (↑) | $P_d$ (↑) | $M_d$ (↓) | $P_a$ (↑) | $F_a$ (↓) | F1 − T (↑) |
| A | **0.7579** | **0.7398** | 0.8324 | **0.8694** | **0.8505** | **0.9659** | **0.0341** | 0.8773 | 0.1227 | 0.9194 |
| B1 | 0.6795 | 0.6932 | 0.8015 | 0.8117 | 0.8066 | 0.9388 | 0.0612 | **0.9394** | **0.0616** | **0.9391** |
| B2 | 0.7243 | 0.6902 | 0.8317 | 0.7889 | 0.8097 | 0.9516 | 0.0484 | 0.7597 | 0.2413 | 0.8449 |
| C1 | 0.7348 | 0.7278 | **0.8684** | 0.8149 | 0.8408 | 0.9431 | 0.0569 | 0.8328 | 0.1672 | 0.8845 |
| C2 | 0.6691 | 0.6725 | 0.8026 | 0.7935 | 0.7980 | 0.9459 | 0.0541 | 0.6081 | 0.3919 | 0.7403 |

Compared with the pixel-level metrics results of A, B1, and B2 in Table 9, the U-FE module obtained the best results. The U-FE module performs more convolution operations on shallow features and the feature maps at shallow layers provide more detailed information, which makes U-FE perform better than FPN. The U-FE module performs two downsampling operations and the receptive field of the convolution kernel is larger and more global features are extracted, which makes U-FE perform better than FCN. The two sets of experiments show that both shallow and deep features are important for small target detection. More downsampling will drown the features of small targets in the deep network. Without downsampling, the convolution receptive field will be limited and semantic feature information will be missing. Therefore, in order to obtain a better detection effect, it is necessary to select a suitable number of downsampling operations.

Compared with the pixel-level metrics results of A, C1, and C2 in Table 9, the SPSA module obtained the best results. Compared to the SPSA, C1 using TSA decreased by about 2% and C2 using CONV decreased by about 7%. The superiority of the SPSA over TSA indicates the following. Firstly, the scaling down operation in multi-head self-attention does not result in performance loss. Secondly, the features of different regions of interest are mutually interfering information, and calculating self-attention locally in the region of interest can avoid interference from redundant features. The sharp decline of CONV indicates that compared with the convolution of the local receptive field, the global receptive field of self-attention can better excavate the internal correlation between features.

Comparing the results of the target-level metrics in Table 9, A achieves the optimum for $P_d$ and $M_d$ and sub-optimum for $P_a$, $F_a$, and F1 − T. There is an interesting experimental result: the B1 network using FPN is poor in pixel-level metrics, but it is optimal in target-level metrics, $P_a$ and $F_a$. We analyze that this may be related to the operation of fusing multi-layer features simultaneously in FPN.

*5.5. Comparison with SOTA Methods*

To demonstrate the superiority of the proposed CCRANet, we performed a comparison to four other methods: the low-rank-based method IPI [14] and three deep learning methods—ALCNet [19], DNANet [7], and IAANet [24]. The metrics results of the contrast experiments are shown in Table 10.

As can be seen from Table 10, compared with the model-based method IPI, deep learning methods have better ability to cope with complex and diverse infrared images. In deep learning methods, the CCRANet achieves optimality in almost all metrics, and the improvement is obviously relative to the suboptimal network. In terms of pixel-level metrics, the CCRANet outperforms the suboptimal DNANet by about 5% on both nIoU and IoU and by about 3% on F1 − P. In terms of target-level metrics, compared to the second-best DNANet, the CCRANet improves $P_d$ by about 6%, $P_a$ by about 1%, and F1 − T by about 3%.

**Table 10.** Comparison of pixel-level and target-level metrics for the contrast experiments. The best result values are marked in **bold**.

| Method | Pixel-Level Metric | | | | | Target-Level Metric | | | | |
|---|---|---|---|---|---|---|---|---|---|---|
| | nIoU (↑) | IoU (↑) | R (↑) | P (↑) | F1 − P (↑) | $P_d$ (↑) | $M_d$ (↓) | $P_a$ (↑) | $F_a$ (↓) | F1 − T (↑) |
| IPI | 0.1986 | 0.1536 | 0.1689 | 0.6301 | 0.2663 | 0.5846 | 0.4154 | 0.4014 | 0.5986 | 0.4760 |
| ALCNet | 0.6621 | 0.7077 | 0.8150 | 0.8432 | 0.8288 | 0.8990 | 0.1010 | 0.8393 | 0.1607 | 0.8681 |
| DNANet | 0.6916 | 0.7076 | 0.7751 | **0.8905** | 0.8288 | 0.9047 | 0.0953 | 0.8583 | 0.1417 | 0.8809 |
| IAANet | 0.6724 | 0.6451 | 0.7763 | 0.7925 | 0.7843 | 0.7696 | 0.2304 | 0.6051 | 0.3949 | 0.6775 |
| CCRANet | **0.7579** | **0.7398** | **0.8324** | 0.8694 | **0.8505** | **0.9659** | **0.0341** | **0.8773** | **0.1227** | **0.9194** |

In order to comprehensively measure the pixel-level prediction ability of different deep learning methods, here the PR curve and ROC curve of the CCRANet and the other three methods (see Figure 8) were plotted. As seen in Figure 8, the CCRANet PR curve can cover the PR curve of other methods. This shows that the CCRANet's prediction accuracy is stronger for positive samples (i.e., the pixels corresponding to the target region in the image). After the ROC curve stabilized, the recall metric of CCRANet is better, and it is always better than the other networks under different false-alarm rates.

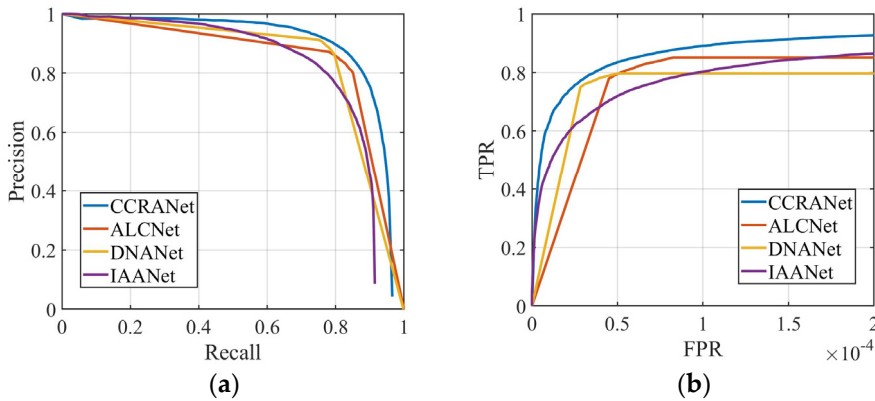

**Figure 8.** The PR curve and ROC curve of the CCRANet and contrast deep learning methods. (**a**) PR curve. (**b**) ROC curve.

The model processing speeds of the CCRANet and other three deep learning methods are compared in Table 11. The average speed of different methods on the test set of the SLR-IRST was calculated using the same data computing platform. As seen in Table 11, the CCRANet is slower than ALCNet in model processing but faster than the other two methods. Considering the improvement of the CCRANet in terms of prediction indicators, it can be considered that the CCRANet also has advantages in model processing speed.

**Table 11.** Comparison of model processing speeds between the CCRANet and contrast networks.

| Method | ALCNet | DNANet | IAANet | CCRANet |
|---|---|---|---|---|
| Fps (↑) | 111.09 | 45.68 | 24.43 | 66.52 |

We undertook a comprehensive comparison of pixel-level metrics, target-level metrics, and the results of model processing speed. It showed that the CCRANet can stably obtain a better infrared small target detection performance in the SLR-IRST dataset compared with comparison methods.

In order to observe the prediction abilities of different networks more clearly and intuitively, the visual prediction results are shown in Figure 9. Three scenarios are selected randomly from the test dataset, as shown in Figrue 9a. The corresponding labels are shown in Figure 9b. The visual prediction results of the CCRANet and other four methods are presented in Figure 9c–g. In Figure 9, the green, red, and yellow circles represent correctly

detected targets, missed targets, and false targets, respectively. The red box in the lower left corner is a partial enlargement of the small target.

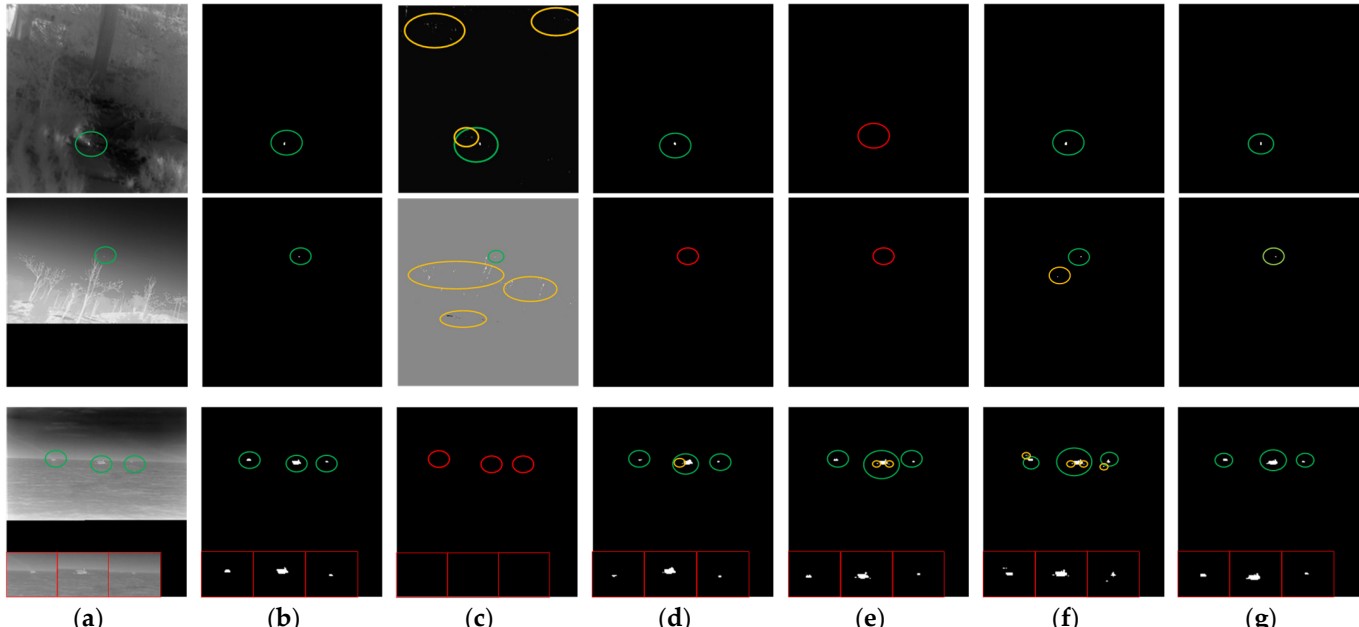

**Figure 9.** Visualization of test images for the SLR-IRST dataset. (**a**) image. (**b**) label. (**c**) IPI. (**d**) ALCNet. (**e**) DNANet. (**f**) IAANet. (**g**) CCRANet.

In order to further evaluate the prediction performance of the CCRANet for low-resolution infrared small target images, we collected some real infrared small target images. The backgrounds of the collected infrared images include the sky, clouds, buildings, trees, tower cranes, rivers, etc. Acquisition angles include looking up and looking down. Small targets in the infrared images include small drones, birds, and other creatures. These images and those in the SLR-IRST were acquired using different devices in different scenarios with data from different sources, and the correlation is low.

The single-frame collected infrared small target images, and the dataset (SAC-IRST) with a resolution of 256 × 256 was obtained after the filling and scaling of the images. It can further evaluate the generalization ability of different network training models. There are 80 images in the SAC-IRST, including 40 infrared images with targets and 40 infrared images without targets. There is interference information such as small targets in images without targets. This can be used to evaluate the ability of different methods to suppress false alarms. The comparative experimental results are shown in Tables 12 and 13.

**Table 12.** Comparison of pixel-level and target-level metrics of different methods on the SAC-IRST dataset (with target images). The best result values are marked in **bold**.

| Method | Pixel-Level Metric | | | | | Target-Level Metric | | | | |
|---|---|---|---|---|---|---|---|---|---|---|
| | nIoU (↑) | IoU (↑) | R (↑) | P (↑) | F1 − P (↑) | $P_d$ (↑) | $M_d$ (↓) | $P_a$ (↑) | $F_a$ (↓) | F1 − T (↑) |
| IPI | 0.1907 | 0.1439 | 0.2165 | 0.3003 | 0.2516 | 0.8043 | 0.1957 | 0.2681 | 0.7319 | 0.4022 |
| ALCNet | 0.5178 | 0.4886 | 0.5584 | 0.7963 | 0.6565 | 0.8478 | 0.1522 | 0.7800 | 0.2200 | 0.8125 |
| DNANet | 0.5331 | 0.5398 | 0.5866 | **0.8714** | 0.7012 | 0.9348 | 0.0652 | 0.8600 | 0.1400 | 0.8958 |
| IAANet | 0.5350 | 0.4673 | 0.5108 | 0.8459 | 0.6370 | 0.9348 | 0.0652 | 0.7818 | 0.2182 | 0.8515 |
| CCRANet | **0.5555** | **0.5832** | **0.6450** | 0.8588 | **0.7367** | **0.9565** | **0.0435** | **0.8800** | **0.1200** | **0.9167** |

**Table 13.** Comparison of false-alarm number (Fa Num) of different methods on the SAC-IRST dataset (without target images). The best result values are marked in **bold**.

| Method | IPI | ALCNet | DNANet | IAANet | CCRANet |
|--------|-----|--------|--------|--------|---------|
| Fa Num($\downarrow$) | 256 | 16 | 14 | 62 | **10** |

As can be seen in Tables 12 and 13, the CCRANet has a significant advantage over other methods in terms of images with targets. The CCRANet also shows better suppression of false alarms due to untargeted images. The above experiments show that the CCRANet has good generalization ability.

In order to observe the prediction performance of different methods on actual acquired images more clearly, two images with targets and two images without targets are randomly selected (see Figure 10).

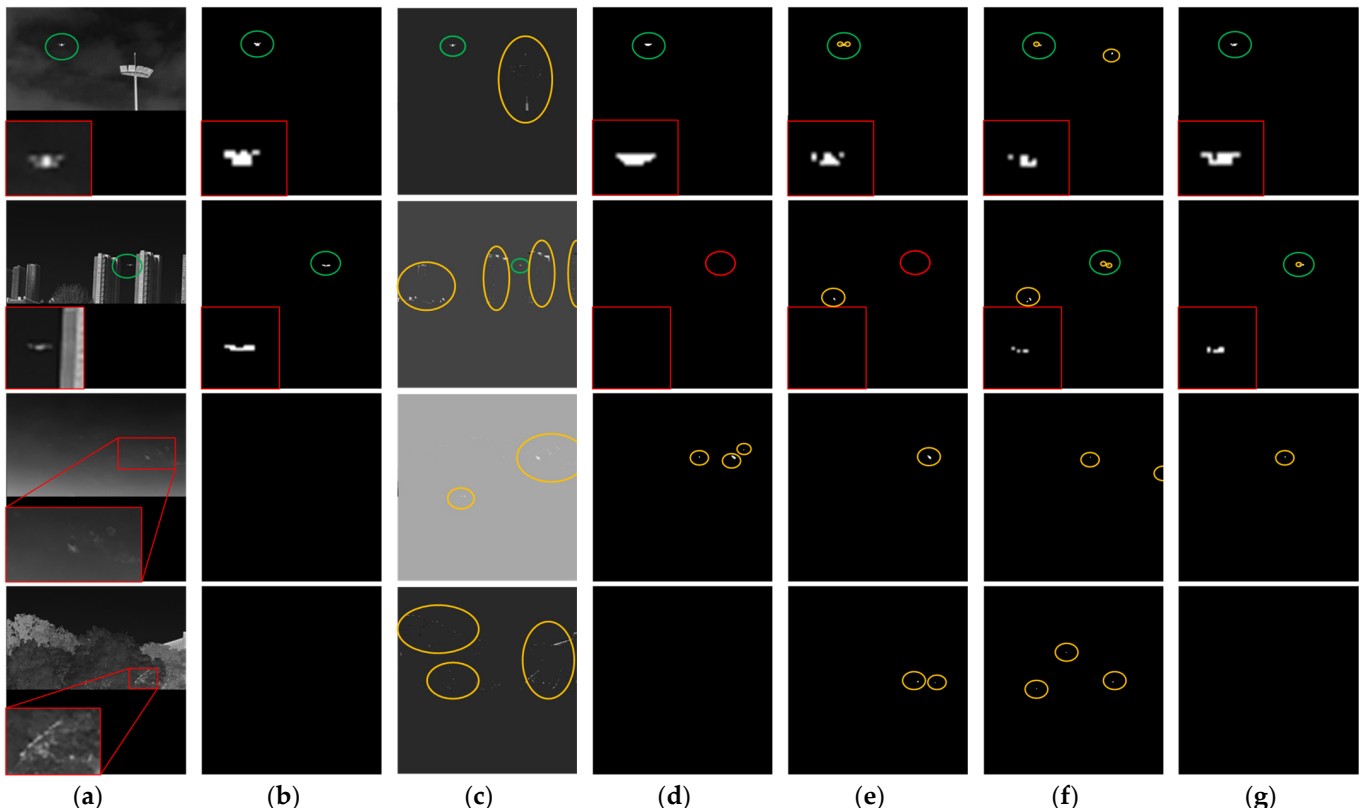

**Figure 10.** Visualization of test images for the SAC-IRST dataset (the top two pairs are images with targets and the bottom two are images without targets). (**a**) image. (**b**) label. (**c**) IPI. (**d**) ALCNet. (**e**) DNANet. (**f**) IAANet. (**g**) CCRANet.

As can be seen from the above results, infrared images from different sources present new challenges to all methods. Our proposed CCRANet still performs better in the SAC-IRST dataset due to its excellent network structure. However, false alarms still exist. As shown by the third line in Figure 10, false alarms are still not avoided for some target-like point clouds. Additionally, the first line in Figure 10 shows that the prediction of targets in uneven grayscale is incomplete. The improvement of the prediction integrity of gray uneven targets is a direction that needs to be considered later.

## 6. Conclusions

In this paper, we constructed a high-quality and large low-resolution infrared small target dataset by sorting, improving, and expanding existing datasets. At the same time,

based on the data characteristics of low-resolution infrared small target images, a coarse-to-fine two-stage target detection method, CCRANet, is constructed. We fully considered the small, weak, and sparse characteristics of infrared small targets and designed a local self-attention network structure with shared parameters, which improves the prediction results of the method and reduces the number of parameters and calculations undertaken by the network. To better evaluate the target-level prediction results of the methods used for infrared small targets, we also proposed two sets of complete target-level evaluation metrics to measure the target-level prediction abilities of the methods in an all-around way.

For the CCRANet method, there are still some directions for further research. In the CCRS module, only the regions of interest were reserved for further feature extraction and prediction. This hard decision method increased the influence of the CCRS module prediction performance on network results. Therefore, in future research, a soft decision method should be tried, the weight of the interest regions should be increased, and the weight of the non-interest regions should be reduced rather than simply discarded. In addition, the time information in the multi-frame sequence should be introduced to further enhance the accuracy of network prediction.

**Author Contributions:** Conceptualization, W.W. and C.X.; methodology, simulation and validation, W.W., C.X. and H.D.; data curation, W.W., G.Z., Y.H. and Z.C.; writing—original draft preparation, W.W.; writing—review and editing, W.W., C.X., H.D., R.L. and H.Y. All authors have read and agreed to the published version of the manuscript.

**Funding:** This research was funded in part by the Qian Xuesen Youth Innovation Fund 2020-QXSQNCXJJ-01, and in part by the Young Star of Science and Technology in Shaanxi Province 2023KJXX-104.

**Data Availability Statement:** The SIRST, IRSTD-1k, IRST640, SIRST-v2, IRDST, and sea ship infrared image data used to support the research is available from the websites https://github.com/YimianDai/sirst, accessed on 29 July 2020; https://github.com/RuiZhang97/ISNet, accessed on 20 March 2022; https://github.com/jzchenriver/IRST640, accessed on 17 August 2022; https://github.com/YimianDai/open-sirst-v2, accessed on 21 February 2023; http://xzbai.buaa.edu.cn/datasets.html, accessed on 2 April 2023; and http://iray.iraytek.com:7813/apply/Sea_shipping.html/, accessed on 10 December 2021.

**Acknowledgments:** The authors would like to thank D.Y., Z.M., C.G. and S.H. for providing the data. And the authors would like to thank InfiRay for providing the data, too.

**Conflicts of Interest:** The authors declare no conflict of interest.

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
