# Peer review of "CCRANet: A Two-Stage Local Attention Network for Single-Frame Low-Resolution Infrared Small Target Detection"

_remotesensing, doi:10.3390/rs15235539_

Round 1

Reviewer 1 Report

Comments and Suggestions for Authors

See Attachment

Comments on the Quality of English Language

Overall, the English language is decent and sufficient for reading.

Reviewer 2 Report

Comments and Suggestions for Authors

A very well presented paper. However, since low-slow small target detection and classification is a hot-spot in recent years, more state-of-the-art research works which are of significance in this filed should be introduced in Section One.

Comments on the Quality of English Language

Minor editing of English language required

Reviewer 3 Report

Comments and Suggestions for Authors

11/10/2023

Dear authors,

In the manuscript CCRANet: A Two-Stage Local Attention Network for Single-Frame Low-Resolution Infrared Small Target Detection you combine the target characteristics of low esolution infrared small target images and studies the infrared small target detection method under complex background based on attention mechanism.

General comments

The introduction is too short, and you have used very few references.

Such manuscripts should be written in the third person. You have mentioned the word 'we' 64 times in the text. By the way, this greatly irritates the reader. So, please change this throughout the text.

I see confusion in the writing as the main flaw of this text. The dataset should be better defined (not through references) and how all these slopes affect the originality of the image. Do they cause false alarms. Because, when interpreting a picture, it is important that the picture is as original as possible to ensure that it shows the real state of affairs.

Specific comments (are in the manuscript)

-          Lines 28-49 – You state a lot of facts without references!

-          Line 50 - Such type of manuscript should be written in the third person. So, please change this throughout the text.

-          Lines 66-70 - You state a lot of facts without references!

-          Line 145 - Delete, excess.

-          Lines 149-157 - Again, you are stating a lot of facts without references.

Best regards

Reviewer 4 Report

Comments and Suggestions for Authors

Dear Authors,
Thank you for your work related to small target identification.
I see you define a small target at line 71, but this definition is not precise.

" Dot targets occupy less than 10 pixels in the whole image, and the size of the target is about 3×3, with no specific shape information, resembling a circular dot from the outside."

You use images having various resolutions (see Table 3). Your definition should take into account the image resolution. It is quite different to have a small target of 10 pixels in an image 640x512 with respect to 96x135. Your definition can be used because you detailed the dataset and provided the Data Availability Statement.

Please define exactly what you mean by "small target".

Best Regards

Reviewer 5 Report

Comments and Suggestions for Authors

The authors delve into the pivotal domain of infrared small target detection technology, underscoring its broad applications in fields like infrared search, tracking, and precision guidance, among others. This paper offers a commendable contribution in the form of a newly curated dataset named the single-frame low-resolution infrared small target (SLR-IRST) dataset. This dataset not only stands as a testament to the authors' efforts in expanding and sorting existing datasets but also represents a thorough evaluation based on various parameters such as target quantity, categorization, and size. Alongside the dataset, the research introduces a two-stage small target detection network. This network, tailored specifically for enhancing the pixel-level metrics of low-resolution infrared small target detection, exhibits superior performance compared to existing methods, especially when benchmarked on the SLR-IRST dataset. Not only does it outshine in pixel and target-level metrics, but it also showcases a competitive edge in processing speed. A noteworthy initiative by the authors is the decision to release both the dataset and the detection network on Github, which undoubtedly facilitates transparency, collaboration, and the potential for iterative improvements by the broader research community. However, a minor shortfall of the paper is the lack of a comprehensive conclusion detailing prospective future work. Addressing this would provide clearer direction and context for subsequent researchers in the field.

Comments on the Quality of English Language

Minor editing of English language required

Reviewer 6 Report

Comments and Suggestions for Authors

The paper proposes a method for small target detection of low-resolution infrared images and construct a dataset called SLR-IRST to improve the detection ability of DL networks. The contributions are significant, and the writing is well structured. However, some issues should also be addressed before it can be published.

1, pls clarify whether the detection targets are moving targets. If the network detects all small objects whether they are moving all not, pls emphasize the ability of the network that it can supress ignore background pixels and simultaneously concentrate on interested pixels; If the network only detect moving targets, pls explain why it can distinguish moving targets from non-moving small targets.

2, it is very confused for me that how do you define small objects, by their physical sizes or pixel sizes? It is known to us all that the pixel size of a small targe will have larger pixel size if the camera focal length is great or the target is close to the camera. Since in a tracking task, we expect a constant id of a target regardless of pixel size (as in the abstract, you say that small object detection is widely used in search and tracking). In my opinion, the early alarm may be your ultimate purpose.

3, I am not very sure how can you get the diameters of circular regions of interests in CCRS module, since the output only includes the spatial and confidence information.

4, pls explain why U-FE module is used and what role does it play. 

5, Since x has a size of k×pi×l×l×D (as shown in figure 6), how can it expand to k×2l×2l×D, may a diagram is needed.

6, Also in Figure 6, in generating the attention, why the cross product is operated on original feature map (i.e., x) and the split feature map, this operation is very different from original proposed attention mechanism (i.e., method in paper, Attention is All You Need). There needs an ablation experiment to compare your trick and the original method.

Comments on the Quality of English Language

 The writing is well structured, while a minor revision is still needed since there are some flaws.

Round 2

Reviewer 1 Report

Comments and Suggestions for Authors

The authors have addressed all my comments. The paper can be accepted as is.

Reviewer 3 Report

Comments and Suggestions for Authors

11/11/2023

Dear authors,

In the manuscript CCRANet: A Two-Stage Local Attention Network for Single-Frame Low-Resolution Infrared Small Target Detection you combine the target characteristics of low resolution infrared small target images and studies the infrared small target detection method under complex background based on attention mechanism.

General comments

You answered all my comments and questions. I am mostly satisfied with them and have no further comments.

Best regards

Reviewer 6 Report

Comments and Suggestions for Authors

The Authors answer all my questions and address my concerns.